# From Catheter Complication to Surgical Success: Urgent Retrieval of an Embolized Amplatzer Device and Valve Repair

**DOI:** 10.3390/reports8030185

**Published:** 2025-09-19

**Authors:** Iulia Raluca Munteanu, Ramona Cristina Novaconi, Adrian Petru Merce, Lucian Silviu Falnita, Ciprian Nicușor Dima, Horea Bogdan Feier

**Affiliations:** 1Doctoral School Medicine-Pharmacy, “Victor Babes” University of Medicine and Pharmacy Timisoara, Eftimie Murgu Square No. 2, 300041 Timisoara, Romania; iulia.munteanu@umft.ro; 2Clinic of Cardiovascular Surgery, Institute of Cardiovascular Diseases of Timisoara, Gheorghe Adam Street, No. 13A, 300310 Timisoara, Romania; adimerce@gmail.com (A.P.M.); lfalnita@gmail.com (L.S.F.); dima.ciprian@umft.ro (C.N.D.); horea.feier@umft.ro (H.B.F.); 3Advanced Research Center of the Institute for Cardiovascular Diseases, 300310 Timisoara, Romania; 4Department VI Cardiology-Cardiovascular Surgery Clinic, “Victor Babes” University of Medicine and Pharmacy Timisoara, Eftimie Murgu Square No. 2, 300041 Timisoara, Romania

**Keywords:** atrial septal defect, Amplatzer device, device embolization

## Abstract

**Background and Clinical Significance:** Atrial septal defects (ASDs), particularly the ostium secundum type, are congenital cardiac anomalies that can lead to serious complications if left untreated. Percutaneous closure using devices like the Amplatzer Septal Occluder (ASO) has become a widely accepted approach, although complications such as device embolization can occur. **Case Presentation:** We present a unique case of a 28-year-old woman who developed acute hemodynamic instability and arrhythmias following embolization of an Amplatzer device into the right ventricle during an ASD closure. Despite initial treatment with antiarrhythmic medication, the patient required urgent open-heart surgery for device retrieval and ASD closure. The surgery successfully involved pericardial patch closure of the ASD, device removal from the right ventricle, and the performance of the Kay procedure to address significant tricuspid regurgitation. Postoperative recovery was uneventful, with the patient stabilized and discharged in stable condition. **Conclusions:** This case highlights the critical need for rapid surgical intervention in cases of device embolization, and the importance of multidisciplinary coordination in managing such complex complications. The combination of ASD closure, device retrieval, and tricuspid valve repair led to a successful outcome, underscoring the importance of timely, decisive action in complex cardiovascular emergencies.

## 1. Introduction and Clinical Significance

Secundum atrial septal defects (ASDs) are among the most common congenital cardiac anomalies in adults, frequently resulting in chronic left-to-right shunting that can lead to right ventricular (RV) volume overload, atrial dilation, arrhythmias, and ultimately pulmonary hypertension and heart failure if left uncorrected [1,2,3,4].

Percutaneous closure using occluder devices—most notably the Amplatzer Septal Occluder—has become the first-line intervention for anatomically suitable defects, owing to its less invasive nature, shorter hospital stay, and lower risk of early complications compared to open-heart surgery [5,6,7].

Patient selection for transcatheter ASD closure relies on a comprehensive assessment of shunt significance (e.g., Qp:Qs ≥ 1.5), RV dilation, pulmonary arterial pressure, and adequacy of septal rims, as per AHA/ACC and ESC guidelines [1,4,8,9,10]. Contraindications include complex anatomies (primum or sinus venosus defects), inadequate septal tissue, or elevated pulmonary vascular resistance refractory to medical therapy [3,8,9].

Despite its overall safety, transcatheter closure carries a recognized risk of device embolization, reported in approximately 0.4–1.1% of cases. Risk factors include large defect size, deficient or floppy rims, undersized or malpositioned devices, and procedural technical errors [1,2,3,4,6,11]. Retrieval is often feasible percutaneously if embolization occurs early and is anatomically accessible; however, embolization into the RV or other less accessible chambers typically necessitates urgent surgical retrieval [2].

The current case is noteworthy due to the rare and urgent scenario it presents. A device embolized immediately into the RV outlet tract, precipitating acute hemodynamic and electrical instability. This necessitated emergent surgical retrieval combined with ASD closure and tricuspid valve repair. Reporting such a case is crucial for multiple reasons: it underscores the importance of meticulous preprocedure imaging and decision-making algorithms; it highlights the necessity of multidisciplinary surgical backup in ASD closure programs; and it documents an infrequently observed combination of complications and surgical solutions in clinical practice. These aspects make the case both rare and highly instructive for the field.

## 2. Case Presentation

A 28-year-old woman arrived at the emergency department intubated and sedated following complications during a percutaneous atrial septal defect closure procedure using an Amplatzer device. No prior clinical data were available at the time of emergency transfer. While the absence of documentation initially limited our understanding of the procedural events, subsequent efforts to retrieve data from the referring institution were unsuccessful. This limitation reflects the real-world challenges of managing high-risk cardiac transfers and underlines the importance of structured inter-institutional communication. In our institution this is the first case encountered in over 32 years of experience, underscoring its rarity but potential severity. The patient’s medical history was unknown, as no prior clinical data were available. During the procedure, the Amplatzer device embolized into the right ventricle. Upon arrival, she was in critical condition, with low blood pressure (90/60 mmHg), tachycardia, and electrical signs of cardiac distress on the electrocardiogram (ECG), including short episodes of ventricular tachycardia, bigeminism, and trigeminism. Antiarrhythmic medication (Xylocaine) was administered, but there was no significant improvement in her condition.

Given the acute hemodynamic instability and the risk of further embolization, an urgent decision was made to proceed with open-heart surgery for device retrieval and ASD closure. The patient was taken to the operating room, where a median sternotomy was performed, and cardiopulmonary bypass was established. Transesophageal echocardiography performed during the procedure revealed that the Amplatzer device (Figure 1a) was located in the right ventricular cavity. Intraoperative TEE estimated the maximal ASD (Figure 1b) diameter at approximately 35–38 mm. Balloon sizing was not performed, as the patient was already undergoing open-heart surgery and device reimplantation was not considered. Given the context of emergent surgical retrieval, detailed rim assessment was not undertaken. The right atrium was accessed through the right atrial appendage, and the ASD was successfully closed with an autologous pericardial patch, effectively sealing the communication between the left and right atria (confirmed by TEE—no residual shunt).

Once the ASD was repaired, attention was directed toward retrieving the embolized Amplatzer device from the right ventricle. Under direct visualization, the device was carefully removed without causing additional damage to the right ventricular wall or other structures. Although the device was removed without visible trauma to the right ventricular endocardium or tricuspid apparatus, intraoperative testing revealed moderate to severe tricuspid regurgitation. This may have been caused by transient mechanical interference from the embolized device, leading to annular dilation or leaflet malcoaptation. Similar functional regurgitation has been reported in cases of right heart volume overload or device-related irritation. After confirming the device was fully retrieved, the tricuspid valve was assessed for signs of dysfunction, particularly any regurgitation that may have been exacerbated by the embolization event.

To evaluate tricuspid valve function, saline solution was instilled into the right ventricle, which revealed moderate to severe tricuspid regurgitation. The saline infusion test was corroborated by intraoperative TEE, which confirmed moderate-to-severe tricuspid regurgitation. Given the extent of the regurgitation, a tricuspid valve repair was indicated. The Kay procedure, a technique involving posterior annuloplasty and leaflet plication, was performed to optimize valve coaptation and reduce regurgitant flow. This method is particularly useful when annular dilation is present without severe leaflet pathology. Intraoperative TEE showed mild to moderate tricuspid regurgitation post-procedure.

Postoperatively, the patient was transferred to the intensive care unit for monitoring. Following the surgical intervention, her hemodynamics stabilized, with resolution of the arrhythmias. Postoperative echocardiography confirmed the complete closure of the ASD, retrieval of the device, and an improvement in tricuspid valve function. The patient was stable, awake and extubated just 4 h postoperatively, with no additional cardiovascular support needed for optimal hemodynamics.

The patient showed progressive improvement over the next few days and was discharged after 7 days, in stable condition, after an uneventful recovery. At the 3-month follow-up, the patient was asymptomatic, with normal echocardiographic findings, no evidence of residual shunt, and preserved tricuspid valve function. She reported no arrhythmias or exercise intolerance.

The combination of ASD closure, device retrieval, and the Kay procedure successfully addressed the patient’s immediate and long-term cardiac needs, leading to a favorable outcome. This case exemplifies the importance of timely and decisive action in the face of acute cardiovascular emergencies. The combination of ASD closure, device retrieval, and tricuspid valve repair resulted in a successful recovery with no long-term sequelae. The patient’s outcome underscores the critical role of multidisciplinary coordination and the need for rapid surgical intervention in complex cases of percutaneous device complications.

**Figure 1 reports-08-00185-f001:**
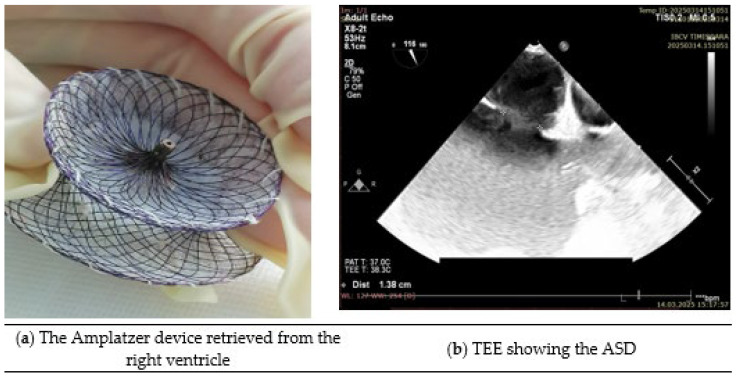
(**a**) The Amplatzer device retrieved during surgery. (**b**) Intraoperative transesophageal echocardiography from the same patient, demonstrating the secundum atrial septal defect prior to closure.

## 3. Discussion

Percutaneous closure of atrial septal defects using occluder devices has become a widely accepted alternative to traditional surgical repair. While it offers the advantage of less invasive procedures with reduced recovery times, complications can still occur, with device embolization being one of the more serious events. Device embolization can result in device migration to other chambers of the heart, causing arrhythmias, hemodynamic instability, or even paradoxical embolism [[12],[13],[14],[15],[16],[17],[18],,[19],[20]].

Incidence of embolization of the Amplatzer devices varies in the range of 0.4–1.1% as reported by literature [13,14,15,16,17]. Larger ASDs were associated with a higher risk of embolization [13,14]. Although procedural information from the referring institution was unavailable, several mechanisms could plausibly explain device embolization in this case. Large defects exceeding 36 mm are known to be at higher risk of embolization, particularly in the presence of deficient rims or undersized devices [21]. The intraoperative finding of a defect measuring 35–38 mm suggests that borderline size alone may have contributed to instability of the implanted occluder. Lack of detailed rim characterization precludes definitive attribution, but deficient posteroinferior support or technical factors at deployment remain likely contributors [6,15,21]. This finding suggests that the size of the ASD and device selection are critical factors in minimizing procedural complications [16]. Multiple risk factors for device embolization have been reported in the literature, including deficient or floppy rims, oversizing or undersizing of the device relative to defect diameter, and operator inexperience [6,15,21]. In our case, although detailed procedural information was unavailable from the referring center, the large defect size (35–38 mm) and possible rim deficiency likely contributed to device instability and migration

Most embolizations are reported to occur acutely, either during the index procedure or within the first 24–48 h, although late events have also been described [19,20]. In our case, embolization occurred immediately during deployment, underscoring the importance of intra-procedural vigilance and preparedness for emergency management [15,22,23].

When device embolization occurs, immediate recognition and intervention are vital. In cases where percutaneous retrieval fails or is not possible, surgical intervention remains a reliable option [15,16,17,19,20]. As demonstrated in our case, the Amplatzer device that embolized into the right ventricle was successfully retrieved during open-heart surgery. The ASD was closed using a pericardial patch, and tricuspid valve regurgitation was assessed by saline instillation, which revealed moderate to severe regurgitation. The Kay procedure was then performed to address the valve insufficiency, demonstrating a favorable outcome. This approach was selected owing to the operating surgeon’s experience with favorable long-term outcomes in his practice, and because it allows a shorter operative time than ring annuloplasty—an important consideration given the patient’s critical condition.

Narayanan et al. [16] also highlighted the risk of embolization in patients with large ASDs and suggested that the selection of occluders must take into account the anatomical characteristics of the defect to avoid complications. Although previous reports have documented Amplatzer device embolization, few have detailed cases requiring urgent surgical retrieval from the right ventricle with concomitant tricuspid valve repair. Our case contributes to this limited body of literature by demonstrating a multidisciplinary approach and highlighting the need for individualized surgical planning in high-risk cases. Preventive strategies are essential to avoid device embolization. These include detailed preprocedural imaging with transesophageal echocardiography and multi-slice computed tomography (MSCT) to assess the size, shape, and surrounding rims of the ASD. Intraprocedural balloon sizing helps confirm defect diameter, and a “tug test” is routinely performed before device release to ensure stability. Lack of adherence to these steps may significantly increase embolization risk, especially in large or irregular defects. Furthermore, early detection is crucial, as delays in diagnosis may lead to worsened outcomes [16,17,18].

Several case series have documented embolization of Amplatzer devices into the right ventricle, though most could be retrieved percutaneously [17,20]. Surgical retrieval is required in a minority of cases, particularly when the device lodges in the right ventricular outflow tract or when percutaneous techniques fail [5,15,24]. Reports of concomitant tricuspid valve repair in this setting are exceedingly rare, highlighting the uniqueness of our case and the value of sharing our institutional experience.

In terms of management, surgical intervention remains a vital part of the strategy when percutaneous methods fail. The retrieval of embolized devices, combined with the closure of the ASD and any necessary valve interventions, generally leads to favorable outcomes [[15],[16],[17],[18],,[19],[20]]. Garre et al. [15] found that patients who underwent surgical retrieval of embolized devices had no further complications, with a mean hospital stay of five days and no recurrence of defects during one-year follow-up.

Rather than a routine retrieval, this case illustrates the interplay of multiple high-risk features: a large defect, immediate embolization, and device impaction at the tricuspid valve with resultant regurgitation. The combined strategy of urgent retrieval, ASD patch closure, and Kay annuloplasty underscores the importance of individualized surgical planning and rapid multidisciplinary action in rare but life-threatening complications.

## 4. Conclusions

Percutaneous ASD closure is generally safe, but device embolization—particularly into the right ventricle—remains a rare, life-threatening complication that requires immediate recognition and surgical readiness. Our case illustrates how the combination of urgent device retrieval, surgical closure of a large defect, and concomitant tricuspid valve repair can lead to favorable outcomes, even in critically unstable patients. Beyond the technical aspects, this case highlights the importance of structured referral pathways: the absence of procedural documentation from the referring center hindered timely assessment, emphasizing the need for standardized inter-hospital communication protocols in cardiovascular emergencies. By reporting this rare scenario, we aim to reinforce the necessity of comprehensive preprocedural imaging, surgical backup for percutaneous programs, and improved information transfer between institutions.

## Data Availability

The original contributions presented in this study are included in the article/Appendix A. Further inquiries can be directed to the corresponding author.

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
