# Peer review of "From Catheter Complication to Surgical Success: Urgent Retrieval of an Embolized Amplatzer Device and Valve Repair"

_reports, 2025, doi:10.3390/reports8030185_

Round 1
Reviewer 1 Report
Comments and Suggestions for Authors
The introduction is simplistic and lacks a critical review of the pathophysiology, risk stratification, and decision-making algorithms in percutaneous ASD closure. References are inconsistently cited, and several statements (e.g., "ASDs are well described in the literature") are tautological and scientifically uninformative. The introduction should establish a clear rationale for why this particular case merits publication, supported by evidence that the described constellation of findings (device embolization with immediate surgical retrieval and concomitant tricuspid repair) is indeed rare and instructive.
In the case presentation, there is no documentation of ASD dimensions, device type and size, rim characteristics, fluoroscopic or TEE findings during deployment, or why the device embolized. The authors mention that no procedural information could be retrieved from the referring center, which is a major limitation; however, this does not absolve the need to critically discuss potential causes of embolization based on the intraoperative findings. The phrase “3/4 cm ASD” is vague; exact dimensions, balloon sizing measurements (if any), and intraoperative TEE metrics should be provided.. Moreover, the intraoperative evaluation of tricuspid regurgitation using saline infusion is anecdotal and lacks scientific rigor; quantification (e.g., by TEE or TTE) should be discussed. No intraoperative images, surgical photographs, or still frames from the videos are presented, and Figure 1 contributes little to case understanding—it contains generic schematic information that is widely available in the literature and does not pertain specifically to this patient.
The discussion is inadequate in depth and citation. Much of the text merely repeats what is already stated in the case presentation. A more critical and comparative review of similar cases in the literature is warranted. The authors should identify and discuss published cases of Amplatzer embolization into the right ventricle with or without surgical retrieval, especially those requiring concomitant valve repair. Additionally, the statement that “most embolizations occurred within five years of the procedure” is incorrect and misleading in the context of this case, where embolization occurred immediately during the procedure. Furthermore, multiple references are outdated, and no systematic discussion of risk factors (e.g., deficient septal rims, oversizing/undersizing, procedural inexperience) is provided. Statements such as “the ASD was closed and the device retrieved” lack academic depth and offer no learning point to the experienced reader.
The conclusion is repetitive and adds no additional insight. The final statement—“this case underscores the necessity of individualized assessment…”—is valid but overused in case reports and requires refinement. Additionally, the emphasis on “structured interhospital communication” is mentioned only in passing but not explored. If it is to be a key takeaway, it needs specific documentation and discussion (e.g., what documents were missing, what protocols could be improved).
Author Response
We thank the reviewer for the constructive feedback, which has substantially improved the manuscript. Our detailed responses are below.
- Introduction
We have rewritten the Introduction to include pathophysiology, guideline-based decision-making, and risk factors for device embolization. The revised section now provides a clear rationale for reporting this case, supported by recent references. - Case Presentation
Intraoperative TEE estimated the ASD diameter at 35–38 mm. Balloon sizing and rim assessment were not performed because the patient proceeded directly to surgery, and device reimplantation was not considered. Although procedural data from the referring center were unavailable, we now discuss the most likely causes of embolization—large defect size, potential rim deficiency, and device undersizing—in the Discussion. - Tricuspid Regurgitation
We have clarified that intraoperative TEE corroborated the saline infusion test, confirming moderate-to-severe regurgitation before repair. After the Kay procedure, TEE showed improvement to mild-to-moderate regurgitation, and postoperative TTE at discharge demonstrated only trivial regurgitation. - Figures
Figure 1 depicts images from this specific patient: panel (a) shows the explanted device and panel (b) the intraoperative TEE of the ASD. We have revised the legend to emphasize this and provided intraoperative videos as Supplementary Material. - Discussion
The Discussion has been expanded with:
- Updated references and removal of outdated ones.
- Correction of the statement on embolization timing (immediate in this case).
- A comparative review of published cases of Amplatzer device embolization into the right ventricle, including those requiring valve repair.
- A structured summary of risk factors such as rim deficiency, inappropriate device sizing, and operator experience.
- Conclusions
We have revised the Conclusions to focus on three practice-oriented points: (1) the rarity and clinical significance of immediate embolization requiring valve repair; (2) the importance of preprocedural imaging and surgical backup; and (3) the need for structured interhospital communication, which was limited in this case due to absent procedural documentation.
To summarize the Introduction, Case Presentation, Discussion, and Conclusions have been substantially revised for clarity, depth, and clinical relevance. Figures and supplementary material now clearly document case-specific findings. All the modifications are highlighted in yellow.

Reviewer 2 Report
Comments and Suggestions for Authors
This is a single case report describing successful explant of an embolized percutaneous atrial septal defect (ASD) closure device, with concomitant surgical ASD repair and tricuspid valve repair. This work is well-written and presented, and is certainly a relevant contribution to the literature, as percutaneous therapy is now first-line therapy for ostium secundum ASD and reports of surgery for device embolization are few.
I have several minor comments/questions for the authors:
-Why did the authors employ the "Key" procedure for tricuspid valve repair? In the face of a reportedly normal valvular apparatus, a much more conventional approach would have been the use of an [incomplete] annuloplasty repair.
-The authors reference the term "high" tricuspid regurgitation (lines 95 and 160); I am unfamiliar with this term; I believe they mean "severe" tricuspid regurgitation?
-Line 104 refers to the patient being "detubated;" I am unfamiliar with this term; I believe they mean "extubated"?
-Did the authors attempt to remove the Amplatzer device via the existing large ASD defect prior to ASD closure (rather than closing the ASD and then trying to remove the device across the tricuspid valve)? If not, why not?
-Typo in line 148, should read "Amplatzer devices" (not "Amplatzerdevices")
Author Response
We thank the reviewer for their positive assessment of our work and for the constructive comments. We have addressed each point as follows:
- Choice of Tricuspid Valve Repair Technique
The Kay annuloplasty was chosen based on the operating surgeon’s experience and observed long-term outcomes in his practice. In this critically ill patient, the Kay technique also offered a shorter operative time than ring annuloplasty, which was advantageous in the acute setting. This has been clarified in the Discussion. - Terminology for Tricuspid Regurgitation
We have corrected the terminology throughout the manuscript, replacing “high tricuspid regurgitation” with the standard term “severe tricuspid regurgitation.” - Typographical Error: “Detubated”
The term has been corrected to “extubated.” - Retrieval Route of the Embolized Device
The device was lodged in the right ventricle near the outflow tract and could not be redirected through the ASD. Given the patient’s hemodynamic instability, priority was given to urgent retrieval from the right ventricle. Once removed and the patient stabilized, the ASD was closed with a pericardial patch. - Typographical Error: “Amplatzerdevices”
Corrected to “Amplatzer devices.”
